# Genomics and Epigenomics of Gestational Diabetes Mellitus: Understanding the Molecular Pathways of the Disease Pathogenesis

**DOI:** 10.3390/ijms23073514

**Published:** 2022-03-23

**Authors:** Nadia Abu Samra, Herbert F. Jelinek, Habiba Alsafar, Farah Asghar, Muhieddine Seoud, Shahad M. Hussein, Hisham M. Mubarak, Siddiq Anwar, Mashal Memon, Nariman Afify, Ridda Manzoor, Zahrah Al-Homedi, Wael Osman

**Affiliations:** 1Department of Biology, College of Arts and Sciences, Khalifa University, Abu Dhabi P.O. Box 127788, United Arab Emirates; nadija.samra@ku.ac.ae; 2Department of Biomedical Engineering, Khalifa University, Abu Dhabi P.O. Box 127788, United Arab Emirates; herbert.jelinek@ku.ac.ae (H.F.J.); habiba.alsafar@ku.ac.ae (H.A.); 3Health Innovation Engineering Center, Khalifa University, Abu Dhabi P.O. Box 127788, United Arab Emirates; 4Center for Biotechnology, Khalifa University, Abu Dhabi P.O. Box 127788, United Arab Emirates; 5Department of Molecular Biology and Genetics, College of Medicine and Health Sciences, Khalifa University, Abu Dhabi P.O. Box 127788, United Arab Emirates; 6Department of Obstetrics and Gynecology, Sheikh Shakhbout Medical City, Abu Dhabi P.O. Box 11001, United Arab Emirates; fasghar@ssmc.ae (F.A.); mikeseoud@ssmc.ae (M.S.); smmahmoud@ssmc.ae (S.M.H.); 7Department of Obstetrics and Gynecology, Danat Al Emarat Hospital for Women and Children, Abu Dhabi, United Arab Emirates; hisham.mubarak@danatalemarat.ae; 8Department of Fetal Medicine, Danat Al Emarat Hospital for Women and Children, Abu Dhabi, United Arab Emirates; 9Department of Nephrology, Sheikh Shakhbout Medical City, Abu Dhabi P.O. Box 11001, United Arab Emirates; sanwar@ssmc.ae; 10College of Medicine and Health Sciences, Khalifa University, Abu Dhabi P.O. Box 127788, United Arab Emirates; 100049880@ku.ac.ae (M.M.); 100049875@ku.ac.ae (N.A.); 100052897@ku.ac.ae (R.M.); 100049881@ku.ac.ae (Z.A.-H.)

**Keywords:** pregnancy, gestation diabetes mellitus, GDM, hyperglycemia, epigenetic, epigenetic modifications

## Abstract

One of the most common complications during pregnancy is gestational diabetes mellitus (GDM), hyperglycemia that occurs for the first time during pregnancy. The condition is multifactorial, caused by an interaction between genetic, epigenetic, and environmental factors. However, the underlying mechanisms responsible for its pathogenesis remain elusive. Moreover, in contrast to several common metabolic disorders, molecular research in GDM is lagging. It is important to recognize that GDM is still commonly diagnosed during the second trimester of pregnancy using the oral glucose tolerance test (OGGT), at a time when both a fetal and maternal pathophysiology is already present, demonstrating the increased blood glucose levels associated with exacerbated insulin resistance. Therefore, early detection of metabolic changes and associated epigenetic and genetic factors that can lead to an improved prediction of adverse pregnancy outcomes and future cardio-metabolic pathologies in GDM women and their children is imperative. Several genomic and epigenetic approaches have been used to identify the genes, genetic variants, metabolic pathways, and epigenetic modifications involved in GDM to determine its etiology. In this article, we explore these factors as well as how their functional effects may contribute to immediate and future pathologies in women with GDM and their offspring from birth to adulthood. We also discuss how these approaches contribute to the changes in different molecular pathways that contribute to the GDM pathogenesis, with a special focus on the development of insulin resistance.

## 1. Introduction

In a healthy pregnancy, a woman’s body adapts to the developing fetus through physiological and anatomical changes [1]. This is manifested by weight gain, placental development, and changes in the metabolic system as well as almost all organ systems [1,2]. An important metabolic change that occurs during pregnancy is the gradual reduction in insulin sensitivity [1,3]. As a result, glucose is absorbed and stored at early stages of pregnancy for the energy requirements required later in gestation [1]. Nevertheless, during the second half of pregnancy (20–24 weeks onward), increased insulin resistance is observed. This is suggested to be mediated via placental hormones including: human placental growth (hPG), human placental lactogen (hPL), cortisol, estrogen, and progesterone, in addition to chronic inflammation factors like tumor necrosis factor α (TNF-α) [1,2,4]. As a result, the pancreatic beta-cells undergo hypertrophy and hyperfunction to produce more insulin, and thus compensate for the maternal insulin resistance [1,5]. During some pregnancies, this may lead either to dysfunction of the β-cells due to excessive secretion of insulin or insufficient compensation of the blood glucose load, or both, which will ultimately result in gestational diabetes mellitus (GDM) [1,5,6].

The prevalence of GDM, which is a form of hyperglycemia that first develops during pregnancy [1], increases worldwide every year, with prevalence rates varying by ethnicity [7]. According to the International Diabetes Federation (IDF) Diabetes Atlas for 2021, the prevalence of GDM (20–49 years old) varies from 13% in Africa to 25.9% in Southeast Asia (SEA) [8]. Additionally, hyperglycemia in pregnancy (HIP) affects approximately 16.7% of all pregnancies worldwide, 80.3% of which is caused by gestational diabetes mellitus [8]. Obesity, type 2 diabetes mellitus (T2DM), previous history of GDM, advanced maternal age (35 years old or older), and polycystic ovary syndrome (PCOS) have all been identified as risk factors for GDM [1] (Figure 1).

Several acute GDM complications may arise during pregnancy or delivery such as pre-eclampsia (a high blood pressure disorder), macrosomia (a large fetus), and shoulder dystocia [1,9,10]. In addition, although GDM resolves after delivery, it can still have long-lasting effects on both the mother and fetus [1]. Several studies have indicated that mothers are seven times more likely to develop T2DM a few years after their first diagnosis of GDM, in addition to an increased risk of cancer, cardiovascular disease, and kidney disease [11,12]. The offspring of mothers with GDM are at higher risk of being obese when they reach adulthood, developing type 2 diabetes, and having impaired neurocognitive development [12]. This leads to a vicious cycle of obesity and diabetes between generations [1].

Specific details regarding the mechanism by which this condition develops and progresses are still lacking. Reports suggest that GDM is a multifactorial disorder, and that in addition to genetic factors, environmental factors such as tobacco smoke, physical inactivity, and alcohol consumption, as well as medications and lifestyle factors, are likely to contribute to its development [13]. Consequently, these factors may also have an impact on the growing fetus by altering the epigenome, resulting in epigenetic programming that may expose the fetus to chronic diseases such as obesity and type 2 diabetes in the future [14]. Research on clinical, genomic, and molecular biology is continuously focusing on defining the pathophysiology and molecular architecture of GDM. There is currently a focus on identifying genetic variants, the molecular pathways involved, and expression profiles of the epigenetic modifications. To do this, technological advancements are necessary. In this paper, we present a review of the genetics and epigenetics of GDM, as well as the genomic strategies used to collect this information.

A genetic linkage map that demonstrates the relative locations of all GDM-related genes reviewed in this paper is illustrated in Figure 2. The genes were investigated and classified according to the technology used to identify them and based on whether they are epigenetically modified. GDM-related genes were identified on all chromosomes except for chromosomes 13 and 17 and the sex chromosomes X and Y (Figure 2). 

## 2. Genomic Approaches Used in Understanding Gestational Diabetes

### 2.1. Candidate-Gene Association Studies

To date, candidate-gene studies have been the most widely used method for identifying genetic variants associated with GDM [15]. The reason is that these are inexpensive, quick, and allow scientists to examine the implications of an educated guess regarding the genetic basis of GDM [16]. The candidate-gene approach thereby allows for the validation of an association between a gene and phenotype by assessing the effect of its genetic variant [16,17]. There are a variety of methods for identifying candidate genes, including animal model studies, positional cloning, genome-wide association studies, and next-generation sequencing approaches, and these methods are prioritized according to their biological feasibility and prior knowledge of gene function [15,17,18].

A foundation for selecting suitable candidate genes for GDM has been established since 1964, when O’Sullivan and Mahan found a correlation between gestational diabetes and the development of diabetes several years later [19]. Since then, molecular studies have shed light on the relationship between T2DM and GDM, and have demonstrated important similarities between the pathophysiology of both disorders such as increased insulin resistance or a limited ability of the β-cell to overcome insulin insensitivity [20,21]. Several epidemiological studies have also demonstrated a correlation between the prevalence of GDM and T2DM [22,23,24]. Furthermore, there are indications of familial clustering and heritability of T2DM and GDM [25,26]. Considering all of these observations, it is not surprising that both disorders may have a similar genetic basis. Therefore, T2DM susceptibility genes, as well as those that are involved in β-cell function, insulin response, and glucose regulation, have been identified as potential candidates for GDM [27,28]. Yahaya et al. suggested that 83 candidate genes exist for GDM in a recent comprehensive review. Among them, *TCF7L2*, *KCNQ1*, *CDKAL1*, *IRS1*, and *MTNR1B* are the most frequently studied for their correlation with GDM [27] (Table 1).

#### 2.1.1. *TCF7L2*

Transcription factor 7 like 2 (TCF7L2) is located on chromosome 10q25 [28]. It is highly expressed in adipose tissue, the pancreas, and several other tissues [53]. *TCF7L2* is the strongest association with T2DM yet, and plays a role in glucose homeostasis [54]. TCF7L2 also regulates adipokines, signaling hormones secreted by adipocytes such as leptin and adiponectin [28,55]. Those hormones are involved in appetite regulation and energy expenditure [28]. Usually, leptin is elevated in GDM cases and is thought to contribute to macrosomia, while adiponectin is lower and associated with insulin resistance [1]. It is not yet known how TCF7L2 contributes to T2DM, but Chen et al. suggested that TCF7L2 regulates adipocyte development [56]. Another possibility is that TCF7L2 is involved in the PI3K/AKT pathway, which facilitates insulin signaling transduction and glucose homeostasis [57]. To the best of our knowledge, seven genetic variants in *TCF7L2* have been positively associated with GDM (Table 1) [29,30,31,32,33,34,35,36,37,38,39,40,52,54,58,59,60,61,62,63,64]. 

#### 2.1.2. *KCNQ1*

Several genetic mutations and disorders can disrupt the insulin secretion process, resulting in early termination of insulin production and the shortage of insulin seen in patients with T2DM and GDM [44]. Potassium voltage-gated channel subfamily Q member 1 (*KCNQ1*) is a gene on chromosome 11p15 that is expressed in the kidney, brain, heart, and pancreatic islets [28,45]. It contributes to cell polarization and regulation of insulin secretion [44,65]. This candidate gene was also selected based on its known association with T2DM [66]. Accordingly, numerous SNPs have been tested for their association with GDM using the candidate-gene approach, especially in Asian populations (Table 1) [27,37,41,42,43,44,46,66,67,68].

#### 2.1.3. *CDKAL1*

Cyclin-dependent kinase 5 regulatory subunits associated protein 1-like 1 (*CDKAL1*) is located on chromosome 6q22.3 [28]. It is reported to be involved in insulin secretion and β-cell function by negatively regulating CDK5 and inhibiting insulin secretion [28,47,69]. In a case-control study involving 316 controls and 321 cases of GDM [47], rs9295478, rs6935599 and rs7747752 were shown to be linked with increased GDM risk. Moreover, rs7754840 and rs7756992 were included in a meta-analysis that covered five studies in five different populations [48], and they were associated with elevated GDM risk in Asians and Caucasians.

#### 2.1.4. *IRS1*

Insulin receptor substrate-1 (*IRS1*) is located on chromosome 2q36.3. It is an endogenous component of the insulin receptor, found in tissues that are sensitive to insulin and involved in the insulin signaling pathway [28,49,70]. Specifically, the genetic variant rs1801278 (G972R) has been reported to increase insulin insensitivity and disrupt IRS1 function [71]. It has been examined in Saudi Arabians [49], Greeks [30], Scandinavians [50], and Russians [72]. SNP rs7578326 was also investigated in Austrian-Hungarian and Finnish populations [36,73].

#### 2.1.5. *MTNR1B*

Melatonin receptor 1B is a member of the melatonin receptor family [28]. It is found on chromosome 11q14.3, encoded by the *MTNRB1* gene. It is expressed in the pancreatic islets and plays a role in glucose homeostasis during pregnancy via the melatonin signaling pathway [74]. Three SNPs have been reported to be associated with elevated GDM risk (rs10830962, rs10830963, rs1387153) [37,38,49,51,52]. RS10830962 was associated with GDM in the Chinese population [51]. *MTNRB1* polymorphism rs10830963 was found to be associated with GDM in Asians, Caucasians, and Arabs [36,38,49,52], while rs1387153 was a risk factor for GDM in Mexicans, Finnish, Danish, and Arab populations [36,37,38,49]. 

### 2.2. Genome-Wide Association Studies (GWAS)

Genome-wide association studies (GWAS) enable the identification of genetic variants that predict the susceptibility to diseases from an individual’s genome. Over 50,000 associations have been reported between diseases and genetic variants since the first GWAS was published in 2005 [75]. To our knowledge, two GWAS have been conducted for GDM [76,77]. Kwak et al. conducted a two-stage study in Korean women that included 468 GDM cases and 1242 controls in the first stage and 931 GDM cases and 783 controls in the replication study [76]. Consequently, two of the most significant GDM variants, rs7754840 and rs10830962, were identified and located in the intron regions of *CDKAL1* and upstream of *MTNR1B*, respectively [76]. The second study observed 115 controls and 103 cases of GDM in Chinese women [77]. The results identified 23 SNPs associated with GDM [77]. These SNPs identified four genes (*CTIF*, *CDH18*, *PTGIS*, and *SYNPR*) that may be involved in GDM [77]. GO enrichment and KEGG pathway analyses revealed that the *CDH18*, *PTGIS*, and *SYNPR* genes were enriched for or located in glycometabolism pathways [77]. As part of a multi-ethnic GWAS in 2013, Hayes et al. examined glycemic traits, such as fasting glucose, fasting C-peptide, and glucose levels 1 h and 2 h after OGTT [78]. The study found HKDC1 and BACE2 to be associated with glycemic characteristics in pregnancy [78]. HKDC1 is associated with impaired glucose tolerance in old-age pregnant mice whose gene is downregulated by 50%, and is embryonically lethal in HKDC1-KO mice [79]. In addition, higher levels of expression of the gene were associated with higher levels of insulin sensitivity and glucose tolerance [80]. As for BACE2, it was reported to be involved in β-cell function [81]. 

### 2.3. DNA Microarrays

DNA microarrays have revolutionized the genomic research field ever since they were introduced in the mid-1990s [82]. In addition to being used for DNA methylation profiling and genotyping, they can also be applied to determine the expression levels of various genes simultaneously [83]. Research is now using this genomic tool to compare the expression patterns of genes in tissues under different health conditions [84]. For example, it is possible to compare the cells of a patient with those of a healthy control to identify genes and biomarkers involved in the pathogenesis of disease [84].

It was through the application of microarrays to the study of GDM that the first molecular basis for the relationship between placental gene modifications and GDM was established [85]. As illustrated in a study conducted in 2003, placental biopsies from GDM cases were analyzed using microarrays and RT-PCR, which uncovered a modification of 435 genes, 18% of which were associated with inflammatory responses [85]. The results of this study suggest that fetal genes are programmed in an inflammatory environment, which contributes to the development of diseases in adulthood [85]. An investigation of the placental genes in placental samples of patients with GDM was conducted in 2009 [86]. Microarray analysis confirmed by RT-PCR indicated that 66 genes regulating biological functions such as cell activation, apoptosis, and the immune response are altered in GDM patients [86]. Expression profiling of placental human chorionic membrane-derived stem cells (CMSCs) from pregnant women with and without GDM revealed upregulation of 162 genes that were linked to migration ability, growth factor-associated signal transduction, and epithelial development [87]. On the other hand, 269 genes were downregulated, and were related to angiogenesis and cellular metabolism [87]. GDM women had a reduction in the expression of ALDH enzymes (detoxification enzymes), which led to an increase in oxidative stress [87]. The results of this study are in accordance with the findings of elevated oxidative stress levels identified in maternal plasma and placental tissues [88]. Additionally, it explains the mother-fetus complications reported in previous studies [89,90]. Moreover, oxidative stress can alter the expression of glucose transporter type 4 (GLUT4) by disrupting nuclear protein transfer to the insulin-responsive element in the GLUT4 promoter, which in turn, affects glucose transport efficiency [28]. Another study, based on evidence of insulin resistance in omental visceral adipose tissues (OVATs) of patients with GDM, utilized a microarray to compare the gene expression profile of GDM cases and healthy controls in a Chinese population [91]. A total of 450 differentially expressed genes (DEGs) were downregulated and 485 were upregulated [91]. Following the construction of a functional interaction network, it was demonstrated that the following five pathways are associated with GDM: cell adhesion molecules, type 1 diabetes, natural killer cell-mediated cytotoxicity, antigen processing and presentation, and TGF-β signaling [91]. This has contributed to a greater understanding of the mechanisms underlying insulin resistance in OVATs of GDM patients [91]. Another microarray study identified several inflammatory genes that are closely associated with GDM, including *CXCL10*, *HLA*, *CXCL9*, and *PTPRC* [92]. CXCL9 was discovered to be enriched in the cytokine signaling pathway, and it is thought to contribute to the development of GDM through regulation of the inflammatory pathway [92]. Similarly, CXCL10 is thought to be involved in the pathogenesis of GDM through its ability to inhibit pancreatic beta-cell proliferation. This was suggested to happen via (a) binding to CXCR3, or (b) interacting with Toll-like receptor 4 for the constant activation of Jun N-terminal kinases and protein kinase B (Akt), cleavage of p21-activated protein kinase 2, and change of Akt signal from proliferation to apoptosis [92]. As for HLA, it has been demonstrated that it contributes to type 1 diabetes mellitus (T1DM) and can influence the development of type 2 diabetes mellitus [92]. In addition, an increased humoral immune response to HLA-class II antigens was detected in GDM cases, indicating that this may mediate the pathogenesis of the disease [92,93]. It has been suggested that PTPRC (protein tyrosine phosphate receptor 5) negatively regulates insulin signal transduction in diabetic cases and is one of the key genes in GDM [92]. In another two studies that applied microarray techniques, cytochrome P450, family 1, subfamily A, polypeptide 1 (CYP1A1), estrogen receptor 1 (ESR1), fibronectin 1 (FN1), and leptin (LEP) were also found to be critical genes for GDM pathogenesis [92].

### 2.4. Next-Generation Sequencing Approaches (NGS)

Over the past few years, genomic research has made substantial progress. This is largely due to the emergence of next-generation sequencing (NGS) technology [94]. NGS provides a more in-depth analysis of an organism’s genome than conventional sequencing (Sanger sequencing), allowing researchers to locate DNA variations and their function in a shorter time [94,95]. Furthermore, several NGS approaches have been used in recent years to understand the genomic mechanisms of disease.

#### 2.4.1. Whole-Exome Sequencing (WES)

Whole-exome sequencing (WES) is one of the technologies used in next-generation sequencing. Approximately 1% of the genome is comprised of protein-coding regions [96]. To our knowledge, WES has been widely used in clinical and genomic research; however, only one study has used this method to identify genetic variants in GDM cases. A recent study by Nikolai Paul et al. screened 50 non-obese GDM Maltese women, and identified three pathogenic variants, rs201815564, rs37046485, and rs766191969, in the *ABCC8*, *GCK*, and *HNF1A* genes, respectively [97]. Additionally, subjects with these variants reported interrupted fasting glucose levels several times [97]. Nevertheless, replication of these studies is needed considering the limited number of subjects, the ethnic heterogeneity of the Maltese population, and the possibility of variants in intronic regions not covered by WES [97].

#### 2.4.2. Whole-Genome Sequencing (WGS)

Whole-genome sequencing (WGS) allows the sequencing of an entire genome. Unlike WES, it identifies variants in both protein-coding (exons) and non-coding genes (introns), as well as in mitochondrial DNA [98]. 

#### 2.4.3. Targeted NGS

The targeted NGS technique allows investigators to sequence a specific region of a genome for a thorough analysis at a lower cost and less time than other NGS methods [99]. Due to its high coverage, sequencing data are more easily interpreted [99]. Target enrichment is an essential step in this technique, which can be achieved either through hybridization with a probe to capture the region of interest, or by amplifying the region by PCR [99]. A study that investigated “Diabetes Panel” genes in 120 GDM patients revealed 45 different pathogenic variants, mostly found in the *GCK* gene, in 38% of patients [100], whereas oligogenic variants were found in four patients [100]. In another study of a Russian population sample, a custom NGS panel targeting 28 diabetes genes was used to sequence 188 patients, 57 of whom were pre-GDM and 131 of whom had GDM. There were 23 pathogenic variants in 59 out of 188 patients, 18 likely pathogenic variants, and 16 of unknown significance. Both pre-diabetic and diabetic groups carried these variants [101], again mainly in the *GCK* gene.

#### 2.4.4. RNA Sequencing 

RNA sequencing has revolutionized our understanding of the transcriptome [102]. NGS sequences’ cDNA and RNA analysis provides a better and more detailed view of alternative splicing and gene expression [102]. RNA sequencing includes the analysis of messenger RNA (mRNA), microRNA (miRNA), non-coding RNA (ncRNA), long ncRNA (lncRNA), and pre-mRNA [102]. In addition, gene expression quantification is highly dependent on sample purity. Due to the heterogeneity of our body cells, RNA sequencing invokes two techniques to overcome this obstacle: laser capture microdissection and cell purification [102]. RNA sequencing has also overcome cell-to-cell variability with a new method known as single-cell RNA sequencing (scRNA-seq), which allows for the analysis of individual cells [102]. 

RNA sequencing has been applied to GDM patients in several studies. Tao et al. identified 647 differentially expressed lncRNA and mRNA in placental tissues from four GDM cases and three controls [103]. A number of enriched signaling pathways of co-expressed mRNAs, such as the Toll-like receptor, which involves CASP8 and TLR5, and endocytosis have also been identified [103]. A similar study by Wang et al. examined the expression of circular RNA (circRNA) in the placenta of 30 GDM cases and 15 healthy controls [104]. Consequently, 8321 circRNA were identified in the placenta, of which 46 were differentially expressed in GDM cases [104]. GO and KEGG enrichment revealed that these circRNAs contribute to advanced glycation end products-receptor for advanced glycation end products (AGE-RAGE) signaling-mediated diabetic complications [104]. Additionally, a novel circRNA, hsa_circ_0005243, has been identified and found downregulated [104]. In a subsequent study, RT-PCR was used to study hsa_circ_0005243 expression in 20 women with GDM and 20 controls [105]. In vitro experiments investigated its role in cell proliferation and migration, as well as in the production of inflammatory factors. Researchers found an increase in inflammatory factors such as interleukin-6 (IL6) and TNF- α in the placenta and plasma of women with GDM [105]. The knockdown of hsa_circ_0005243 suppressed cell migration and downregulated β -catenin, as well as increased nuclear NF-κB p65 nuclear translocation [105]. Based on these observations, downregulation of hsa_circ_0005243 may be associated with GDM pathogenesis via regulation of β-catenin and NF-κB signal pathways [105]. As part of a very recent study, Yang et al. used scRNA-seq for the first time to create a transcriptomic profile of the placenta of GDM patients [106]. A total of nine cell types were identified, with trophoblasts being the most prevalent, along with five associated markers for each type of cell. Furthermore, the functions of DEGs in the placenta were determined, as well as the characteristics of immune cells in the placenta. This profoundly influenced our understanding of the microenvironment of the mother-fetal interface [106].

## 3. Epigenetic Modifications in GDM

Despite having the same genomic information, gene expression patterns vary between different types of cells [107]. This is due to epigenetic modifications, which change the activity and expression of genes without changing their DNA sequence [4]. These modifications are: DNA methylation, histone modifications, and ncRNAs including miRNA [4]. The environment and lifestyle can induce epigenetic changes, such as pollution, tobacco smoking, obesity, lack of physical activity, and alcohol consumption [108]. Furthermore, exposure to such environmental factors can have a butterfly effect: epigenetic modifications may affect biological mechanisms, contributing to the pathogenesis of complex diseases [109]. Their exposure can also impact the epigenetic patterns of subsequent generations [109]. According to Slupecka-Ziemilska et al., the first 1000 days of life, including the preconception and neonatal stages, can determine a person’s future chronic disease susceptibility [14]. This has been observed in the Dutch Hunger Winter study in the Netherlands, which followed up offspring born to mothers that experienced famine and starvation during their gestation in 1944–1945 [110,111]. The study revealed that adult offspring had higher levels of triglycerides, LDL cholesterol, and weight. A higher death rate was found compared to others born at the same time but not exposed to famine in utero [112,113]. 

It is hypothesized that epigenetic modifications play a role in GDM pathogenesis and exhibit the same effect on exposed offspring [4,114]. Researchers found that GDM-exposed offspring have an increased risk of birth defects [115,116,117]. Pavlinkova et al. demonstrated a disrupted transcriptional profile in mice embryos exposed to GDM [116]. Kappen et al. made a similar observation that transcriptomic profiles vary between GDM-exposed offspring and non-exposed offspring [118]. The evidence indicates impairment in gene regulation; however, more research is needed to fully understand these findings. Furthermore, Elliott et al. believe that epigenetic mechanisms should be implicated in adult offspring of GDM mothers to mediate the manifestation of chronic diseases such as T2DM, including the effect of GDM on adult offspring with T2DM, the effect of GDM on epigenetic mechanisms in different tissues and organs, and the effect of epigenetic mechanisms on future T2DM [119]. 

Here, we have reviewed different studies that investigated the epigenetic modifications caused by exposure to GDM. A schematic graph that summarizes the reviewed epigenetic alterations of GDM-linked genes and their molecular effect is presented in (Figure 3).

### 3.1. DNA Methylation

DNA methylation is the addition of a methyl group from S-adenyl methionine (SAM) to the fifth carbon (C5) position of the cytosine nucleotide to form 5-methylcytosine, which is catalyzed by methyltransferase enzymes (DNMTs) with different activities. DNMT3a and DNMT3b carry out new patterns of DNA methylation during embryogenesis (de novo), and DNMT1 maintains methylation status in cells during DNA synthesis [114,120,121]. Generally, methylation is observed at cytosine sites that are followed by guanine (CpG), but rarely at non-CpG sites (CpA, CpG, or CpT) [120]. Gene expression is regulated by DNA methylation either by preventing transcription factors from binding to DNA or by using proteins that inhibit gene expression [120]. Usually, gene activation is linked to hypomethylation of promoters of genes, whereas gene silencing is associated with hypermethylation [114]. Several studies have been performed to study the DNA methylation profile in GDM [122,123,124,125,126,127,128,129,130,131].

In a study by Cardenas et al., bisulfite conversion was used to measure DNA methylation in the fetal placenta among 448 women at 24–30 weeks of gestation after a 2-h oral glucose tolerance test (OGTT), and then the Infinium Methylation EPIC BeadChip was used to determine its levels [122]. As a result, 2-h post-load glycemia in mothers was correlated with hypomethylation of four CpG sites in *PDE4B.* Additionally, *PDE4B* mediates the release of TNF-α, which was found to be elevated in placental and adipose tissues of obese pregnant women compared to non-obese pregnant women [122,132]. TNF-α is also linked to insulin resistance in GDM [122]. In light of this, *PDE4B* inhibitors have been suggested as therapeutic agents [122]. The authors also reported differential methylation of three CpG sites in three genes: *BLM*, *TNFRSF1B*, and *LDLR* [122]. Concerning the *BLM* functions in DNA homeostasis, *TNFRSF1B* has an apoptotic function and *LDLR* encodes for LDL receptors that function in endocytosis [133].

In another study, an Illumina Infinium Human Methylation 450 BeadChip was used to measure DNA methylation in the cord blood of newborns of GDM mothers as well as non-GDM mothers [123]. Accordingly, 4485 differentially methylated sites were found, of which 2150 were highly methylated and 2335 were less methylated [123]. In the research, 37 CpG sites were detected, which belong to 20 genes and may serve as clinical biomarkers for GDM [123]. Gene pathway enrichment analysis further revealed that the T1DM pathway was the most significant KEGG pathway. Gene ontology (GO) pathway analysis then revealed neuron development and immune major histocompatibility complex (MHC) pathway enrichment [123]. 

Another study compared maternal blood samples to umbilical cord blood samples taken from infants in a small sample size of 16 pregnant women (eight with GDM). A total of 381,869 and 540,036 methylation positions were identified in maternal blood and umbilical cord blood, respectively [124]. Among these, the top 200 loci with significant differences were identified, which belong to 167 genes in cord blood and 151 genes in maternal blood; these were found to be differently methylated in GDM groups compared to non-exposed groups, leading to the suggestion that GDM could have epigenetic effects on both mothers and fetuses [124].

Awamleh et al. profiled 42 cord blood samples and 36 placenta samples for DNA methylation using an Illumina Infinium 450 k array [125]. Consequently, 662 CpG sites were identified in placenta samples and 99 CpG sites in cord blood samples at a difference of more than 5% and an aoas1 *p*-value of <0.01 with regard to confounders [125]. Sites that were common for both sample types were found in *PTPRN2* and *AHRR*. The study suggested that this difference in methylation profiles between the two groups was an adaptive response to gestational insulin insensitivity [125]. 

The DNA methylation patterns of GDM cases differed from healthy controls in other studies, as well. Using an Illumina Human Methylation 450 k DNA Analysis BeadChip and whole human gene expression array, Deng et al. analyzed gene expression and DNA methylation in visceral omental adipose tissues (VOATs) in patients with GDM and compared them to those in controls [126]. A total of 485 highly expressed genes and 450 downregulated genes were identified. Seven genes were found to overlap between DEGs and differentially methylated genes (DMGs), of which five were found to be highly methylated and expressed: *C10orf10*, *HLA-DPB1*, *GSTT1*, *FSTL1*, and *HLA-DRB5*. One gene, *HSPA6*, however, was found less methylated and expressed [126]. Gene *MSLN* was the only one hypermethylated with downregulated transcription [126]. Further pathway analysis revealed that antigen processing and presentation pathways are associated with GDM in Chinese pregnant women’s OVATs [126].

In a similar study by Zhang et al., DNA methylation and gene expression profiles of placenta samples from 32 GDM subjects were screened and compared with 31 healthy controls, using the Illumina Infinium Human Methylation 450 BeadChip for obtaining methylation data and the GeneChip^®^ Human Transcriptome Array 2.0 for determining genes expression profiles [127]. There were 24,572 differential CpG sites within 9339 genes, and 931 DEGs between GDM and control samples [127]. In a KEGG and GO pathway analysis of the 326 genes shared by DEGs and DMGs, metabolism and immune-related terms appeared to be enriched [127]. Subsequent protein-protein interaction analysis identified *Oas1*, *Ppie*, and *Polr2g* as target genes for GDM [127].

In addition, several studies have suggested that GDM may also affect offspring via DNA methylation. A genome-wide methylation analysis of peripheral blood samples from children aged 8 to 12 years old, using the Illumina Infinium Human Methylation 27 BeadChip, identified differentially methylated regions (DMRs), from which genes related to cardiometabolic traits were identified: *PANK1*, *NPR1*, *SCAND1*, and *GJA4*. After gene enrichment analysis of the top 84 genes, the ubiquitin-proteasome system (UPS) emerged as the most enriched biological pathway, involved in apoptosis of beta cells, insulin resistance, lipid metabolism, and inflammation [128]. VCAM-1 levels were also elevated in offspring exposed to GDM in utero compared to non-exposed offspring, which may be associated with elevated methylation of *PYGO1* and *CLN8* genes [128]. Moreover, these results point to an early pathogenesis of cardiometabolic diseases [128]. 

In a very recent study, Wang et al. assessed DNA methylation levels at 337 GDM-related CpG sites using MethylTarget sequencing [129]. Peripheral blood samples were collected at an early pregnancy stage from 80 GDM cases and 80 healthy controls. Results of quantitative analysis revealed that DNA methylation levels differed at 13 CpG sites between the two groups. On the other hand, qualitative analysis identified eight CpG sites that differed between the groups, located in the intron region of *C5orf34* and promoter regions of *RDH 12*, *HAPLN3*, *YAP1*, *DNAJB6*, and *NFATC4*. The hypermethylation of the following two CpG sites was correlated with increased GDM risk: CpG site 68,167,324 located in *RDH 12*, and CpG site 24,837,915 located in the promoter region of *NFATC4* [129].

Researchers also revealed that GDM affects the leptin levels of the newborn [130]. DNA methylation near the leptin promoter was examined along with leptin levels in cord blood samples and the results indicated that hypomethylation is associated with high mean cord blood leptin levels [130]. 

The effect of GDM on obesity and T2DM susceptibility was also examined [131]. DNA methylation in the peripheral blood of children with a mean age of 13 was measured, and 48 CpGs associated with in utero diabetes exposure were detected, as well as decreased insulin secretion, a higher body mass index (BMI), and a higher risk of developing T2DM [131]. Most studies confirming the effect of GDM on future T2DM and obesity in adult offspring are either observational, as in the Pima Indian population, where offspring exposed to GDM in utero developed obesity [134], or in animal models such as that by Zhu et al., who found that changes in DNA methylation in the pancreas of mice increased the risk of T2DM [135]. When authors isolate samples from humans, they often concentrate on maternal peripheral blood, cord blood, and the placenta [14]. According to Slupecka-Ziemilska et al., this may not accurately reflect the methylation levels in other metabolism-related organs, such as the pancreas, kidneys, and liver, where DNA methylation levels vary, which needs to be further investigated [14].

### 3.2. Histone Modification

Histones are positively charged proteins that are enriched with lysine and arginine residues, found in eukaryotes [136,137,138]. Their charge allows DNA to easily bind and wrap itself around them, which gives it structural support and facilitates its packaging into the cell nucleus [139]. Histones usually exist as octamers consisting of two each of H2A, H2B, H3, and H4 [137,140]. Each octamer contains a central tetramer composed of two H3 and two H4, linked to dimers of H2A and H2B on either side [137]. Furthermore, histones, just like any other protein, undergo post-translational modifications (PTM), including acetylation, methylation, sumoylation, phosphorylation, and ubiquitylation [140,141]. This gives histones control over gene expression and regulation of chromatin accessibility [141,142]. Of these modifications, acetylation and methylation on lysine residues have been the most investigated [4]. The following studies have investigated histone changes associated with GDM [143,144]: A case-control study by Hepp et al. assessed the expression of histone 3 lysine 9 acetylation (H3K9ac) and histone 3 lysine 4 trimethylation (H3K4me3) in placental tissues of 40 GDM cases and 40 controls, of which 50% were of male fetuses [143]. The H3K9ac levels were found to be downregulated in GDM placentas, whereas the H3K4me3 levels did not differ significantly between controls and GDM subjects [143]. Another study, conducted in 2016, included 39 subjects divided into four groups (non-diabetic women, women with T2DM diagnosed before gestation, women with GDM and postpartum-T2DM, women with GDM and without postpartum-T2DM) [144]. Three blood samples were collected and tested for histone 3 (H3) dimethylation levels at 30 weeks of gestation, 8–10 weeks after birth, and 20 weeks after birth [144]. A comparison of postpartum-T2DM women with non-diabetic women at 8–10 and 20 weeks postpartum showed H3K27 dimethylation was lower in GDM cases by 50–60%. Additionally, GDM women with postpartum-T2DM had a 75% reduction in H3K4 dimethylation levels compared to those without T2DM 8–10 weeks after giving birth [144].

### 3.3. MicroRNA (miRNA)

MicroRNA (miRNA) is a small, non-coding RNA containing 18–25 nucleotides [145,146,147]. miRNA can bind to the untranslated regions (UTRs) of mRNA transcript and prevent its translation into protein, which gives it a critical role in gene regulation [147]. To date, miRNA has been recognized to modulate several diseases, including GDM [148,149,150,151]. Moreover, it regulates GDM by influencing the production of insulin [148]. 

Using the rat insulinoma cell line (INS-1 cells), Feng et al. investigated the role and function of *miR-33a-5p* in GDM [148]. The authors recruited 12 GDM subjects and 12 healthy controls and evaluated their blood glucose levels and *miR-33a-5p* expression. In GDM cases, hyperglycemia in addition to *miR-33a-5p* upregulation was observed [148]. To characterize the functional role of *miR-33a-5p* upregulation, the proliferation rate of INS-1 cells, as well as insulin production, were studied, and the results revealed a reduction in the cellular growth rate and insulin levels [148]. In addition, *miR-33a-5p* inhibition was also studied in *miR-33a-5p* knockdown INS-1 cells, and interestingly, insulin production and its concentration increased in the presence of high and low glucose, supporting the involvement of *miR-33a-5p* in GDM [148]. The study also suggested that miR-33a-5p targets ATP-binding cassette transporter 1 (*ABCA1*) to regulate the function of INS-1 cells [148]. *ABCA1* was involved in HDL biosynthesis and reverse cholesterol transport [148]. It was also shown that lnc-DANCR (differentiation antagonizing non-protein coding RNA) downregulates *miR-33a-5p*, and that its overexpression disrupts the repression effect *miR-33a-5p* exhibits on INS-1 cells [148].

In light of previous studies showing the role of *miR-195-5p* in insulin insensitivity regulation, with its overexpression in GDM women as compared to healthy controls [152,153], Wang et al. explored *miR-195-5p* for its clinical performance and diagnostic value [153]. Serum samples were collected from GDM cases and healthy controls at week 25 of gestation and screened for *miR-195-5p* expression by RT-PCR. GDM cases showed elevated expression of *miR-195-5p* compared with healthy controls. In the analysis of clinical factors between the two groups, BMI, fasting glucose, and the 1-h and 2-h plasma glucose were all positively correlated with *miR-195-5p* expression. The receiver operating characteristic curve (ROC) was used to determine the diagnostic value of *miR-195-5p* for GDM, and it was found to serve as an exceptional biomarker for GDM diagnosis [153]. 

In a study of 21 GDM pregnant women and 10 healthy pregnant controls, *miR-330-3p* was examined [154]. Plasma samples were analyzed for the expression of miRNAs *miR330-3p*, *miR483-5p*, *miR548-3p*, and *miR532-3p*. The GDM group showed upregulation of the first two miRNAs and downregulation of the last two miRNAs compared to controls [154]. An interesting observation was that a subgroup of patients with GDM had different levels of *miR-330-3p*, namely high GDM-miR-330 and low GDM-miR-330. In contrast to low-GDM-miR-330, high-GDM-miR-330 had more destructive diabetic phenotypes. Furthermore, *miR-330-3p* underwent a deep analysis to reveal its functional role in GDM and evaluate some of its previously reported target genes [155,156]. This study confirmed that *miR-330-3p* targets *CDC42* and *E2F1*, which are involved in the beta-cell function and insulin production [154]. 

Tryggestad et al. showed how miRNA expression in human umbilical vein endothelial cells (HUVECs) mediates the GDM effect on metabolic processes in offspring [157]. Researchers compared miRNA expression in HUVECs and the levels of target protein in placentas of GDM infants with normal controls [157]. To identify miRNAs, microarray profiling was performed in HUVECs, followed by in vitro transfection experiments and in vivo experiments to detect their effects on target protein levels. As a result, seven miRNA *(miR-452-5p*, *miR-30c-5p*, *miR-126-3p*, *miR-let-7g-5p*, *miR-130b-3p*, *miR-148a-3p*, and *miR-let-7a-5p*) were identified, and their expression was increased in HUVECs exposed to GDM compared to controls. The levels of the catalytic subunit of AMP-activated protein kinase *α*1 (*AMPKα1*) were reduced in HUVECs transfected with *miR-130b-3p* and *miR-148a-3p.* In placental tissues of infants of mothers with GDM, *AMPKα1* was also reduced [157]. Tryggestad et al. suggest the decrease in *AMPKα1* abundance due to miRNA overexpression may be responsible for the reduced fat oxidation in one-month-old babies, which can lead to future metabolic disorders in the offspring [157,158]. Reduced *AMPKα1* expression also favors the appearance of T2DM and insulin resistance [157]. 

According to Joshi et al., GDM confers an effect on offspring mediated by gender-specific alterations in miRNA and target-gene expression in the fetus [159]. miRNA was analyzed in the amniotic fluid (AF) of 20 women with and 20 without GDM in the second trimester [159]. The AF of women with GDM was found to be upregulated by *miR-199a-3p*, *miR-1268a*, and *miR-503-5p*. Furthermore, female offspring were found to have higher levels of *miR-885-5p*, *miR-378a-3p*, and *miR-7-1-3p* compared to male offspring, whose levels of *miR199a-3p* were higher. Based on the mRNA targets of miRNAs, integrated pathway analysis (IPA) uncovered 166 pathways, 32 in females and 88 in males. Cell growth, inflammation, stem cell development, and cell cycle regulation appear to be affected by these pathways. Additionally, liver-related toxicological pathways were identified using IPA [159]. Since there are not enough GDM-exposed AF cohorts to validate the result, human fetal hepatocytes (PHFHs) exposed to maternal obesity were investigated with the hypothesis that enriched miRNAs found in AF of GDM women originate from the fetal liver. [159]. De novo lipogenesis genes were upregulated in PHFHs, specifically in males, based on miRNA and target gene analysis. In PHFHs of obesity-exposed female fetuses, *miR-885-5p*, *miR-199-3p*, and *miR-503-5p* were upregulated, which correlated with decreased expression of target genes *ABCA1*, *INSR*, and *PAK4*. Similarly, *miR-1268s* and *miR-7-1-3p* were upregulated. In contrast, there were no significant differences in the expression of miRNA in males, but there was an increased expression of their gene targets, *ABCA1*, *PAK4*, and *INSR*. The study suggested the possible contribution of the identified miRNA to the metabolic diseases in GDM- and obesity-exposed offspring in a sex-specific manner [159]. 

Houshmand-Oeregaard et al. studied the effect of maternal diabetes (including GDM) on miR-15 expression in the skeletal muscles of exposed offspring, and hypothesized that this could account for cardiometabolic disease in offspring exposed to maternal diabetes [160]. Previously, *miR-15* was linked with insulin signaling pathways, impacting insulin secretion and sensitivity [160,161,162]. Using biopsies of adult offspring (26–35 years of age) of women with GDM (O-GDM), or T1DM during pregnancy (O-T1DM), miRNA expression was measured and compared with that of offspring of healthy mothers during pregnancy. Results showed upregulation of *miR-15a* and *miR-15b* in the skeletal muscles of the O-GDM and O-T1DM groups compared to controls. In addition, the maternal 2-h OGTT glucose level was found to be positively correlated with *miR-15a* expression in O-GDM, with regard to confounders [160]. The researchers concluded that maternal diabetes during pregnancy causes increased *miR-15a* and *miR-15b* expression in the skeletal muscles of offspring, which may contribute to the pathogenesis of metabolic disorders [160].

Across 36 women with GDM and 80 controls, two hypotheses were explored: circulating miRNA during early-mid-pregnancy is associated with GDM development, and miRNA-GDM associations may differ depending on the pre-pregnancy BMI or the gender of the offspring [163]. Levels of 10 plasma miRNAs (*miR-126-3p*, *-155-5p*, *-21-3p*, *-146b-5p*, *-210-3p*, *-222-3p*, *-223-3p*, *-517-5p*, *-518a-3p*, and *29a-3p*) were measured for their expression in subjects [163]. *MiR-155-5p* and *miR-21-3p* showed an association with GDM when the gestational age was adjusted for. Furthermore, *miR-21-3p* and *miR-210-3p* correlated positively with GDM only in obese women [163]. There were six miRNAs linked to GDM in women carrying male fetuses: *miR-155-5p*, *-21-3p*, *-146b-5p*, *-223-3p*, *-517-5p*, and *-29a-3p* [163]. This study suggests that circulating miRNA in early-mid-pregnancy is associated with GDM in obese women and women expecting boys, which can help to detect at-risk mothers earlier on [163]. 

Zhao et al. studied serum samples of pregnant women at 16–19 weeks and found that women who developed GDM after the 24th–28th week had reduced *miR29a*, *miR132*, and *miR222* expression compared to women who did not develop GDM [164]. To understand the functional role of *miR-29a*, they tested its inhibition, and found that knocking it down can increase insulin-induced gene 1 (*INSIG1*), and thus, increase phosphoenolpyruvate carboxy kinase2 (PCK2) in hepatic cell lines, which enable gluconeogenesis [164]. Other studies found that *miR222* regulates the cell cycle and *miR-132* modifies hepatic metabolism via P450, while *miR-132* deregulation impairs trophoblast development [133].

Guan et al. investigated *miR-21* in GDM patients and normal pregnancies, and revealed that in GDM cases, *miR-21* is downregulated while its target protein, *PPAR-α*, is upregulated, which prevents cell proliferation and infiltration [165]. It was found in another study that *PPARα* expression could also be regulated by *miR-518d* [166]. Qiu and colleagues demonstrated the upregulation of *miR-518d* in plasma and placentas with GDM in vitro and in vivo, along with the decreased expression of its target protein, *PPARα* [166]. This led to glucose-level imbalance, triggering the nuclear transport process of the NF-κB signaling pathway and phosphorylation of pathway-associated proteins, resulting in an inflammatory response (TNF-α, IL-β, IL-6, and COX2) and progression of GDM [166]. 

Another study investigated the effect of GDM on HUVECs in GDM and healthy pregnancies [167]. The attention was toward *miR-101* and one of its many targets, enhancer of zester homolog-2 (*EZH2*). *EZH2* exists as isoforms (α and β), and it trimethylates lysine 27 of histone 3, leading to the inhibition of gene transcription. Assays were conducted on HUVECs from GDM and healthy controls to determine the migration, apoptosis, and Matrigel function. GDM-HUVECs had a (1) reduced functional capacity, (2) elevated *miR-101* expression, and (3) decreased *EZH2-β* levels and trimethylation of histone H3 on lysine 27. The healthy HUVECs were exposed to normal and high glucose concentrations for two days (48 h) and their expression of *miR-101* and *EZH2* was evaluated. Cells exposed to high glucose concentrations had the same results as GDM-exposed cells. Additionally, to identify the mechanism by which *EZH2* regulates *miR-101*, chromatin immunoprecipitation followed by PCR was performed to identify how *EZH2* governs *miR-101* [167]. The results revealed that both GDM and a high glucose concentration decreased *EZH2* binding to the *miR-101* promoter regions in HUVECs [167]. This led to the conclusion that GDM results in fetal endotheliopathy, which contributes to the appearance of cardiovascular diseases in adulthood [167].

miRNA expression profiling was performed in tissues of placenta with GDM, which revealed that *miR-96* was the most downregulated in GDM samples [168]. Further analysis showed that p21-activated kinase 1 (*PAK1*) was highly expressed in GDM samples [168]. Functional assays showed that overexpression of *miR-96* in high or low glucose concentrations enhances insulin secretion and pancreatic beta-cells’ viability, whereas overexpression of *PAK1* leads to cell apoptosis and impaired function of beta-cells, suggesting that *miR-96* targets *PAK1* and that *miR-96* plays a role in the development of GDM through regulation of *PAK1* and beta-cell functions and viability [168]. 

## 4. Molecular Pathways and Pathophysiology of GDM

### 4.1. Insulin Resistance

Insulin resistance is considered the initiating factor of GDM [169,170]. Studies have confirmed that a low-grade systematic inflammation precedes insulin resistance and could contribute to failure of β–cells; nevertheless, defective insulin signaling pathways in maternal adipose and skeletal muscle tissues amplify insulin resistance [169].

Under normal conditions, the glucose regulation process commences when insulin binds to its corresponding insulin receptor (IR), which results in auto-phosphorylation of its tyrosine residues [171]. This allows IR to phosphorylate insulin receptor substrate 1 (IRS-1) on tyrosine residues, which further triggers the phosphorylation of downstream molecules and induces the phosphatidylinositol 3-kinase (PI3K) signaling transduction cascade [171,172]. PI3K, when activated, results in the conversion of phosphatidylinositol 4,5-bisphosphate (PIP2) to phosphatidylinositol (3,4,5)-triphosphate (PIP3). Consequently, downstream 3-phosphoinositide dependent protein kinase1 (PDK1) is activated, which subsequently activates, among other kinases, Akt, resulting in phosphorylation of its substrate (AS160), which regulates translocation of glucose transporter 4 (GLUT4) to the transmembrane and allows for glucose uptake and regulation of protein and lipid metabolism [171,172] (Figure 4).

In GDM pregnancies, decreased expression levels of the following insulin signaling components: IRS1, PIP3, PIK3, and GLUT4, have been reported [173,174,175]. Furthermore, alternative phosphorylation of IRS1 at serine residues was exhibited in GDM patients, which prevents the PI3K signaling cascade from taking place, and thus, inhibits insulin action [176]. The exact underlying mechanism through which disrupted insulin signaling manifests is yet to be clear; however, emerging evidence is shedding light on inflammatory factors and oxidative stress, suggesting they mediate insulin resistance, leading to GDM manifestation (Figure 5). 

### 4.2. Inflammation

The placenta, skeletal muscle, and adipose tissues all contribute to GMD-accompanied inflammation [170,176]. It was suggested that adipose tissues could be interacting with the placenta and causing inflammation and insulin resistance [170]. Both types of tissues exhibited high expression levels of IL-6, IL-8, TNF-α, resistin, and leptin [170,176]. Evidence showed that IL-6 and TNF-α upregulate leptin, which amplifies inflammation as leptin itself elevates IL-6 and TNF-α levels [170,177]. Furthermore, IL-6, TNF-α, and other pro-inflammatory cytokines such as IL-1β, along with receptor proteins such as receptor advanced glycation end-products (RAGE) and Toll-like receptor (TLR) work together and activate the c-Jun-N-terminal kinase (JNK) and nuclear factor kappa-light-chain-enhancer of activated B cells (NF-kB) pathways, which mediate insulin resistance [170,176]. TNF-α was also reported to be elevated in skeletal muscle tissues, inducing the secretion of IL-6, IL-8, and monocyte chemotactic protein 1 (MCP-1), which may induce insulin resistance [176]. Adding to this, GDM cases were also found to exhibit high levels of chorionic gonadotrophin (CG), a pro-inflammatory hormone produced by the placenta. Evidence demonstrates that CG reduces GLUT4 functionality and disrupts glucose uptake [178]. This was suggested to be mediated by the activation of NF-kB and increased expression of pro-inflammatory cytokines [178].

#### 4.2.1. NF-kB Pathways in Insulin Resistance

Generally, the nuclear factor of the kappa light-chain enhancer of activated B-cells (NF-kB) is bound in the cytoplasm by (IκB-α), which is an inhibitory protein, rendering it inactivated [176,179]. However, when pro-inflammatory cytokines such as resistin, TNF-α, IL-6, and IL-1β are produced by the placenta, and/or adipocytes due to increased lipid deposition, the degradation of IκBα is stimulated [170,176,179]. Consequently, NF-kB is activated and translocated to the nucleus, where it is involved in transcription of genes coding for inflammation and insulin resistance [176].

#### 4.2.2. JNK Pathway in Insulin Resistance

Pro-inflammatory factors induce the activation of the JNK pathway, which phosphorylates the serine residues of IRS1. This will inhibit the phosphorylation of tyrosine residues of the insulin receptor, and hence, prevent the PI3K signaling pathway [170,175].

### 4.3. Oxidative Stress

Oxidative stress manifests due to the failure of cellular antioxidants to sustain a balance with the increased levels of reactive oxygen species (ROS) in the body [180]. In normal pregnancies, a state of oxidative stress exists due to the high energy demand and the presence of placenta, which is rich in mitochondria and generates high levels of ROS [181]. In GDM, the stress is enhanced due to hyperglycemia, which disturbs the electron transport chain in the mitochondria, leading to enhanced production of superoxide anion radicals [182]. Oxidative stress in GDM could also be amplified by the activation of the following pathways: protein kinase C (PKC), polyol, hexosamine, and nicotinamide adenine dinucleotide phosphate (NADPH) oxidase [182,183]. Moreover, oxidative stress contributes to insulin resistance by activating JNK and NF-κB [175].

#### 4.3.1. PKC

In response to hyperglycemia, the glycolysis pathway that is responsible for the breakdown of glucose is upregulated. As a result, when fructose 1:6-bisphosphate (a component of glycolysis pathway) breaks down into glyceraldehyde-3-phosphate (G3P), accumulated levels of G3P elevate the production of diacylglycerol (DAG), which in turn, activates PKC that stimulates NADPH oxidase and results in increased ROS [182,183].

#### 4.3.2. Polyol Pathway

This metabolic pathway consists of two steps that mediate glucose conversion into fructose. At first, glucose is reduced to sorbitol and NADPH is oxidized to NADP+, then second, sorbitol, in turn, is oxidized to fructose, and NADH is made from NAD+. NADPH is known to produce glutathione (GSH), which defends against ROS. In GDM, this pathway is upregulated. NADPH thus becomes deficient, and consequently, so does glutathione (GHS). Reduction of GHS production indicates the promotion of ROS [182,183].

#### 4.3.3. Hexosamine

The hexosamine pathway is a subdivision of glycolysis that converts fructose-6-phosphate to glucosamine-6-phosphate. The latter suppresses the glucose-6-phosphate dehydrogenase (G6PD) enzyme, which is a key enzyme in the pentose phosphate pathway (PPP). The PPP is parallel to glycolysis and generates NADPH [184]. In other words, in a hyperglycemic environment, the hexosamine pathway is activated, resulting in suppression of G6PD, which consequently reduces NADPH, and hence, GHS, which increases ROS and oxidative stress [182,183]. 

#### 4.3.4. NADPH Oxidase

NADPH oxidase is a group of enzymes that work toward generating hydrogen peroxide (H_2_O_2_) and superoxide (O^−^_2_) via the transfer of electrons from NADPH to molecular oxygen. In the hyperglycemic state, NADPH oxidase is activated and oxidative stress occurs as a result [182,183].

## 5. Predictive Biomarkers of GDM

Soluble factors secreted by impaired adipose tissues and placenta in GDM cases can be easily detected and quantified, which means they can serve as GDM predictive markers and enable prospective diagnosis [185]. Five types of first-trimester GDM predictors have been reported: (a) blood glucose markers such as fasting glucose, post-load glucose, and hemoglobin A1C; (b) insulin-resistance markers like fasting insulin and sex hormone-binding globulin (SHBG); (c) inflammatory markers such as C-reactive protein and TNF-α; (d) fat cell-derived markers including adiponectin and leptin; and (e) markers in placenta such as placental exosome, placental growth factor (PLGF), and follistatin-like-3 [186]. Also, according to Ma et al., plasma-glycated CD59 (pGCD59) exhibits great potential to serve as an accurate biomarker for the recognition of GDM in early pregnancy and as a risk assessment for delivering large for gestational age (LGA) infants [187]. Adding to this, 1,5-anhydroglucitol (1,5 AG) is also among the emerging GDM biomarkers [188].

In a recent review, it was suggested that visfatin, omentin, leptin, ficolin-3, and fetuin-A could predict GDM during mid-stage pregnancy, whereas fetuin-B, fibroblast growth factor 21 (FGF-21), and plasminogen activator inhibitor 1 (PAI1) could be predictive of GDM in the third trimester of gestation [185].

## 6. Conclusions

GDM is a serious health problem that affects pregnant women around the world. This can also affect the fetus and determine its health profile in adulthood, so early intervention is crucial. The SNPs, genetic variants, and miRNAs studied here could serve as potential biomarkers for GDM, but are highly influenced by the ethnicity and environment. We suggest assessing the combination of molecular biomarkers with serum protein biomarkers to predict GDM in large-scale studies. Furthermore, as GDM exhibits heterogeneity, personalized medicine and targeted treatment approaches should be considered once a better understanding of GDM pathophysiology is acquired. Furthermore, most studies are focusing on the western and eastern regions though MENA is expected to have the second-highest increase in diabetes (GDM risk factor) by 2045 [8]. Therefore, we encourage more research in this region. Beyond this, in agreement with Slupecka-Ziemilska et al., more studies on metabolic tissues are needed to understand the epigenetic changes in metabolic organs [14].

## Figures and Tables

**Figure 1 ijms-23-03514-f001:**
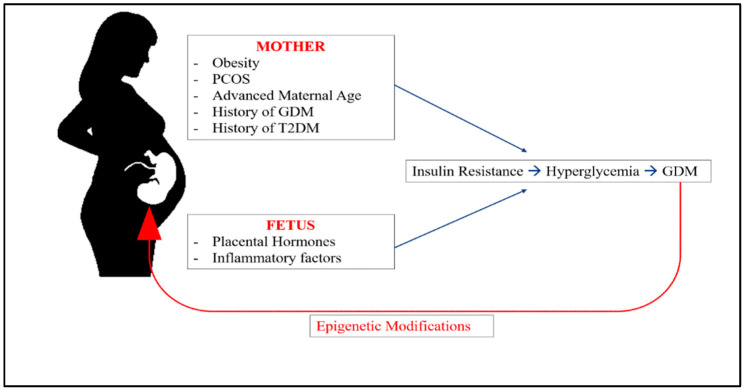
Risk factors for development of GD.

**Figure 2 ijms-23-03514-f002:**
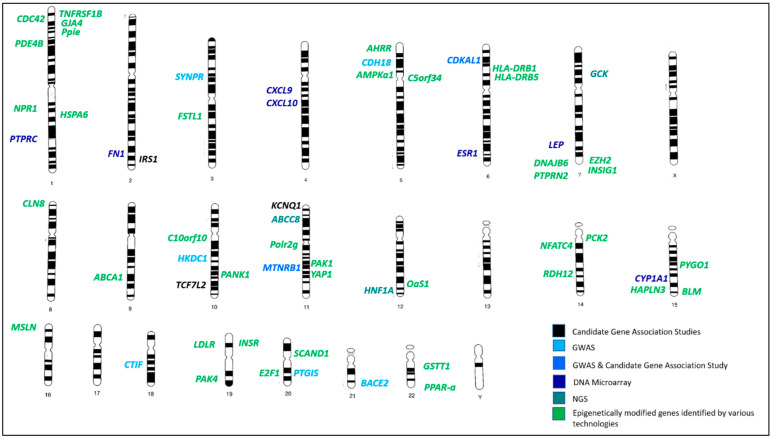
Locations of GDM-linked genes discussed in this review. **Black** color refers to genes identified via candidate-gene association studies; **light blue** color refers to genes identified via genome-wide association studies (GWAS); **blue** color refers to genes identified by both (GWAS and candidate-gene association studies); **dark blue** color refers to genes identified via microarray studies; **dark green** color refers to genes identified via next-generation sequencing technologies (NGS); **green** color refers to genes that have been epigenetically modified and identified by various techniques.

**Figure 3 ijms-23-03514-f003:**
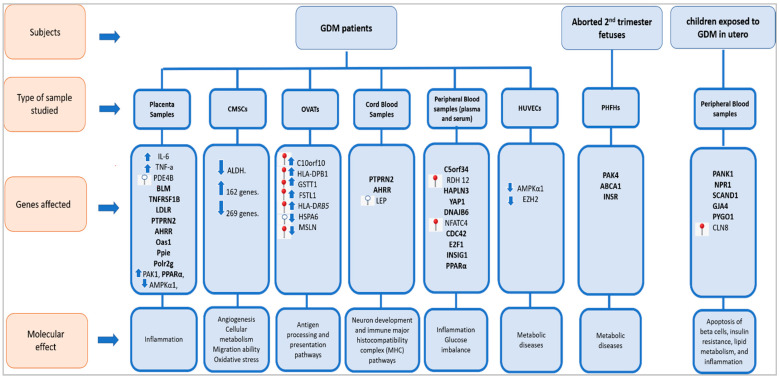
Epigenetic modifications of GDM-linked genes and their molecular effect. Seven types of samples (placental samples; chorionic membrane-derived stem cells (CMSCs); omental visceral adipose tissues (OVATs); cord blood samples; peripheral blood samples; human umbilical vein endothe-lial cells (HUVECs); human fetal hepatocytes (PHFHs)) were isolated from three lines of subjects (GDM patients; aborted second-trimester fetuses; children exposed to GDM in utero) to detect epigenetic modifications and their molecular effect. 
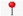
 refers to hypermethylated genes; 
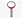
 refers to hypomethylated genes; 
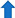
 refers to upregulated genes; 
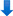
 refers to downregulated genes. Differentially methylated genes are in **Bold**.

**Figure 4 ijms-23-03514-f004:**
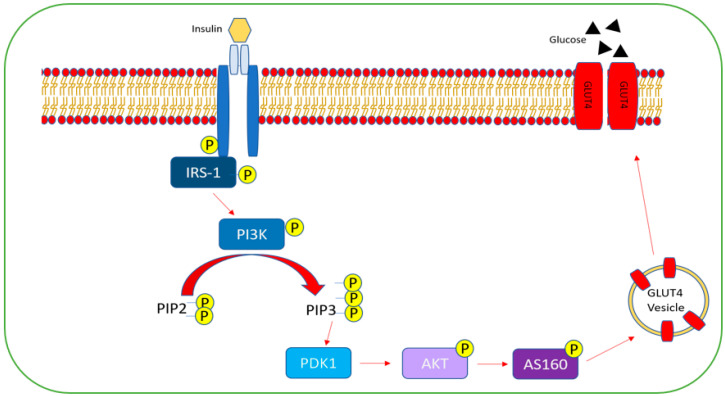
PI3K/Akt signaling pathway. Insulin binds to the insulin receptor, causing autophosphorylation of its tyrosine residues. This causes phosphorylation of insulin receptor substrate-1 (IRS-1) on its tyrosine residues, which leads to the phosphorylation of the phosphatidylinositol 3-kinase (PI3K) signaling transduction cascade. PI3K catalyzes the phosphorylation of phosphatidylinositol 4,5-bisphosphate (PIP2) to phosphatidylinositol (3,4,5)-triphosphate (PIP3). PIP3 activates 3-phosphoinositide-dependent protein kinase-1 (PDK-1) as a result, which in turn, phosphorylates the downstream protein “AKT”, which phosphorylates its substrate AS160. AS160 regulates glucose translocator 4 (GLUT4) and aids in its translocation to the plasma membrane, where it allows glucose to flow.

**Figure 5 ijms-23-03514-f005:**
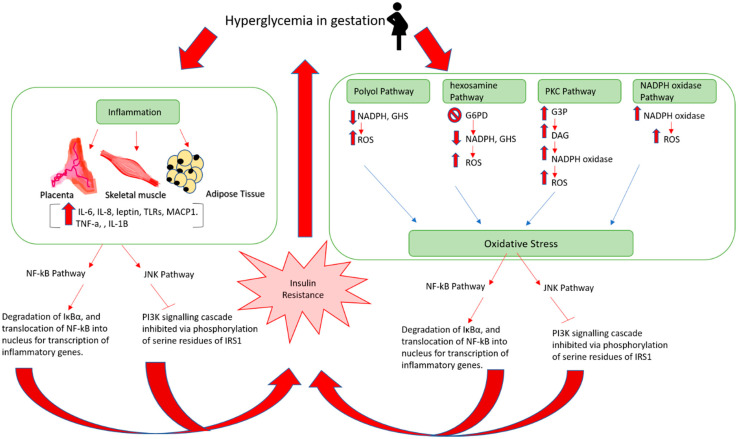
Mechanism of insulin resistance and gestational diabetes. IL-6: interleukin 6; IL-8: interleukin 8; TLRs: Toll-like receptors; MACP1: monocyte chemoattractant protein-1; TNF-α: tumor necrosis factor-α; IL-1β: interleukin 1β; NF-kB: nuclear factor kappa B; JNK: c-Jun N-terminal kinase; NADPH: nicotinamide adenine dinucleotide phosphate; GHS: glutathione; ROS: reactive oxygen species; G6PD: glucose-6-phosphate dehydrogenase; G3P: glyceraldehyde 3-phosphate; DAG: diacylglycerols; PKC: protein kinase C; IRS1: insulin receptor substrate 1.

**Table 1 ijms-23-03514-t001:** Associations of the most investigated GDM genes and their variants across different populations.

Gene	SNP	Population/Ethnicity	Reference
** *TCF7L2* **	rs7903146	Scandinavian	[29]
		Greek	[30]
		Australian and British	[31]
		Danish	[32]
		Korean	[33]
		Swedish	[34]
		Italian	[35]
		Finnish	[36]
		Mexican	[37]
	rs4506565	Mexican	[37]
		Danish	[38]
	rs7901695	American Caucasian	[39]
		Swedish	[34]
		Mexican	[37]
	rs12243326	Mexican	[37]
	rs12255372	Caucasian	[40]
	rs34872471	Danish	[38]
	rs290487	Chinese	[40]
** *KCNQ1* **	rs2237892	Korean	[41,42]
		Chinese	[43]
		Asian	[43]
	rs2074196	Korean	[42]
	rs2237895	Pakistani	[44,45]
		Korean	[41]
	rs2283228	Indian	[46]
*CDKAL1*	rs9295478	Chinese	[47]
	rs6935599	Chinese	[47]
	rs7747752	Chinese	[47]
	rs7754840	Asian and Caucasian	[48]
	rs7756992	Asian and Caucasian	[48]
*IRS1*	rs1801278	Saudi Arabian	[49]
		Greek	[30]
		Scandinavian	[50]
*MTNR1B*	rs10830962	Chinese	[51]
	rs10830963	Asian and Caucasian	[52]
		Danish	[38]
		Finnish	[36]
		Saudi Arabian	[49]
	rs1387153	Danish	[38]
		Saudi Arabian	[49]
		Mexican	[37]
		Finnish	[36]

## Data Availability

Not applicable.

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
