# Peer review of "Genomics and Epigenomics of Gestational Diabetes Mellitus: Understanding the Molecular Pathways of the Disease Pathogenesis"

_ijms, 2022, doi:10.3390/ijms23073514_

Round 1

Reviewer 1 Report

The authors have written a very rigorous paper about the genetic mechanisms underlying gestational diabetes. The title and abstract cover the central aspect of the work. The introduction provides background and information relevant to the study. The scientific content sounds clear is very thoroughly explained. 

Author Response

We would like to extend our gratitude to the reviewer for his/her time and effort in reading the manuscript and his/her positive comments regarding it. Our hope is that this work contributes useful information to the scientific community in regard to understanding this disease pathogenesis, and opens up new possibilities in terms of scientific research. 

Reviewer 2 Report

In this review, the authors tried to unveil the molecular pathways of gestational diabetes mellitus, in terms of genomics and epigenomics. Before the review was suitable for publication, some issues needed addressing.

  1. This review should be improved, and re-edited. Its contents should be well-organized, and understandable.
  2. In lines of 134-136, some sentences should be added in. The authors can explain why these genes were investigated in this review. How were these genes classified? What did X and Y refer for in Figure 3?
  3. Figure 5 should be re-drawn. It could contain more information how inflammation, insulin resistance, and oxidative stress regulate or modulate GDM through NF-κB and JNK pathway, described in this manuscript.

Some minor issues should be addressed:

  1. In Table 1, the same SNP can be placed in the one row. The table would be more readable.
  2. Figure 2 was not essential, so it could be removed. In fact, it was not a figure.
  3. The spelling needed checking in the manuscript. In lines of 860-861, ‘and’ after ‘edited’ should be deleted; In lines of 861-862,’manuscreipt’ should be revised to ’manuscript’.

Author Response

Firstly, we would like to thank the reviewer for the time and effort in reviewing our manuscript. We found these comments valuable and they helped to improve the manuscript. There is also a PDF attachment that contains all of the responses. 

1. This review should be improved, and re-edited. Its contents should be well-organized, and understandable. 

Response:

In response to this point:

  • The review has been re-edited and is now well-organized.
  • Indents have been added to each paragraph
  • Table 1 was put right after the paragraph in which it was mentioned.
  • References were cited in Table 1.
  • In line 212, “s” was added to the word “subunit”
  • In line 240, “s” was removed from the word “enable”
  • In line 306, “- “was added between the words “Toll” and “like”
  • In line 328, “-“was added between the words “ next” and “generation”
  • In line 330, “have” was replaced by “has”
  • In lines 339 and 340, “-“was added between the words “whole” and “genome”
  • In line 361, “analyzes” was replaced by “analysis”
  • In line 378, “s” was added to the word “CircRNA”
  • In line 408, “a” was removed before the word “life”.
  • In line 425, “s” was added to the word “disease”.
  • In line 481, “s” was removed from the word “function”.
  • In line 570, space was removed between the words “consisting” and “of”
  • In line 573, “-“was added between the words “post” and modification”
  • In lines 576, 579, and 718, “lysin” was replaced by “lysine”
  • In line 622, “was” was replaced by “were”
  • In line 752, “s” was added to the word “induce”.

2. In lines of 134-136, some sentences should be added in. The authors can explain why these genes were investigated in this review. How were these genes classified? What did X and Y refer for in Figure 3?

Response:

In response to the reviewer's suggestion, Fig 2 has been removed from the review. Figure 3 has been changed to Figure 2. In lines 123-127, it is explained why the genes have been investigated and how they were classified in Figure 2. X and Y are referred to in the text as "sex chromosomes". Figure 2 has been updated in accordance with the new classification criteria.

3. Figure 5 should be re-drawn. It could contain more information how inflammation, insulin resistance, and oxidative stress regulate or modulate GDM through NF-κB and JNK pathway, described in this manuscript.

Response: 

Thank you for your suggestion. In order to gain a better understanding of insulin/glucose regulation in the state of normoglycemia, Figure 4 has been added to explain that. The second figure, Figure 5, illustrates how hyperglycemia is linked to insulin resistance through two main pathways, and how these pathways influence the pathway followed in the normoglycemia environment illustrated in Figure 4.

The material has been updated from lines 743 to 890. We have added a few more pathways that describe how hyperglycemia leads to oxidative stress, and all the pathways are outlined in Figure 5.

It is our hope that this will provide a better understanding of the role played by these key molecular pathways during GDM pathogenesis.   

Some minor issues should be addressed:

4. In Table 1, the same SNP can be placed in the one row. The table would be more readable.

5. Figure 2 was not essential, so it could be removed. In fact, it was not a figure.

6. The spelling needed checking in the manuscript. In lines of 860-861, ‘and’ after ‘edited’ should be deleted; In lines of 861-862,’manuscreipt’ should be revised to ’manuscript’.

Responses to minor changes:

4. Same SNPs have been put in one row as suggested.

5. Figure 2 has been removed.

6. The spelling was checked in the manuscript, and advised changes were made.

Round 2

Reviewer 2 Report

All my concerns were replied by the authors. I have no further comments on the manuscript.